# The Future of Clinical Active Shoulder Range of Motion Assessment, Best Practice, and Its Challenges: Narrative Review

**DOI:** 10.3390/s25030667

**Published:** 2025-01-23

**Authors:** Wolbert van den Hoorn, Arthur Fabre, Giacomo Nardese, Eric Yung-Sheng Su, Kenneth Cutbush, Ashish Gupta, Graham Kerr

**Affiliations:** 1School of Exercise and Nutrition Sciences, Queensland University of Technology, Brisbane, QLD 4059, Australia; a.fabre@qut.edu.au (A.F.); giacomo.nardese@hdr.qut.edu.au (G.N.); eric.su@qut.edu.au (E.Y.-S.S.); 2School of Health and Rehabilitation Sciences, The University of Queensland, Brisbane, QLD 4072, Australia; 3Queensland Unit for Advanced Shoulder Research, Queensland University of Technology, Brisbane, QLD 4000, Australia; ken@kennethcutbush.com (K.C.); ashish@qoc.com.au (A.G.); 4Australia Shoulder Research Institute, Brisbane, QLD 4000, Australia; 5Faculty of Medicine, The University of Queensland, Brisbane, QLD 4343, Australia; 6Shoulder Surgery QLD Research Institute, Brisbane, QLD 4120, Australia; 7Greenslopes Private Hospital, Brisbane, QLD 4120, Australia

**Keywords:** range of motion, 2D-pose, shoulder, shoulder arthroplasty, narrative review, clinical assessment, accuracy

## Abstract

Optimising outcomes after shoulder interventions requires objective shoulder range of motion (ROM) assessments. This narrative review examines video-based pose technologies and markerless motion capture, focusing on their clinical application for shoulder ROM assessment. Camera pose-based methods offer objective ROM measurements, though the accuracy varies due to the differences in gold standards, anatomical definitions, and deep learning techniques. Despite some biases, the studies report a high consistency, emphasising that methods should not be used interchangeably if they do not agree with each other. Smartphone cameras perform well in capturing 2D planar movements but struggle with that of rotational movements and forward flexion, particularly when thoracic compensations are involved. Proper camera positioning, orientation, and distance are key, highlighting the importance of standardised protocols in mobile phone-based ROM evaluations. Although 3D motion capture, per the International Society of Biomechanics recommendations, remains the gold standard, advancements in LiDAR/depth sensing, smartphone cameras, and deep learning show promise for reliable ROM assessments in clinical settings.

## 1. Introduction

Shoulder osteoarthritis affects approximately one in five individuals over the age of 55 [1], causing persistent and incapacitating pain, primarily during the night, disrupting sleep. This condition not only induces discomfort but also restricts joint movement and impairs upper limb function, significantly compromising various daily activities and the overall quality of life [2]. Shoulder arthroplasty is a viable option for those grappling with severe osteoarthritis, as it aims to enhance shoulder functions and alleviate pain [3]. According to The Australian Orthopaedic Association National Joint Replacement Registry, the prevalence of total joint arthroplasties is on an increasing rise, with >8000 procedures conducted in 2023 alone, with around 71% being a total reverse arthroplasty [4]. Notably, there is a shift in the demographic requiring joint arthroplasty, with a growing proportion of younger individuals (<75 years of age) [4]. Compared to older patients, often, younger patients are more active and expect more function and an earlier return to hobbies and sports. This shift necessitates the development of effective, objective assessments to optimise surgical outcomes and rehabilitation approaches for this evolving patient group, demanding careful objective functional assessment and research.

Since its inception by Paul Grammont [3] in the mid-1980s, the reverse shoulder arthroplasty design has undergone significant evolution, incorporating changes in implant design, positioning, surgical techniques, and planning software. These advancements have contributed to enhanced patient outcomes in terms of function and pain [5,6]. However, there remains room for further improvement in patient outcomes [7]. Considering the evolving landscape of shoulder arthroplasty, with insights into the long-term effects, survivorship [4], and new technological developments on the horizon, such as 3D-navigation, AI and patient-specific instrumentation [6], in combination with higher patient expectations, we need to improve how well we quantify the efficacy of new surgical interventions.

A healthy shoulder range of motion is crucial for everyday tasks, work productivity, sports, and the overall quality of life. Consequently, an active range of motion is identified as the pivotal outcome measure. Despite its significance, there exists a notable absence of a standardised method demonstrating good accuracy, and inter- and intra-rater reliability for assessing a functional shoulder range of motion. Active range of motion is integral to common patient-reported outcome measures. While patient-reported outcome scores aim to establish a standardised approach for quantifying shoulder function and pain, assisting in evaluating treatment efficacy, the methods used to quantify active shoulder range of motion lack standardisation. This inconsistency potentially introduces bias and hampers the ability to compare studies systematically, impeding clinical and research progress in the field. Addressing this gap in methodology standardisation is crucial for advancing research and ensuring unbiased comparisons in systematic reviews assessing the efficacy of surgical interventions.

Various methods are employed to assess the functional shoulder range of motion, ranging from self-assessment [8,9] and visual observation [10,11], using a goniometer, and 2D and 3D motion capture [12]. Figure 1 illustrates the trade-off between processing time (*x*-axis) and measurement accuracy (*y*-axis). Given the time constraints in clinical settings, commonly utilised methods involve the quick quantification of active range of motion, such as visual observation and self-assessment. However, these time-efficient approaches tend to compromise accuracy and reliability [8,11]. Even the clinical gold standard, the universal goniometer, faces challenges with its moderate repeatability [13] and inter-tester reliability [14,15,16]. Striking a balance between rapid assessment and precision is crucial for practical clinical use [17]. The ideal method for clinical application should demand minimal time for assessment while maintaining high levels of accuracy.

Markerless motion capture offers a promising alternative. Recent advances in machine learning algorithms enable the identification of critical body landmarks, simplifying shoulder range of motion estimation. This can be based on a single camera, available with a phone [18]. If clinicians adopt this potentially user-friendly, time-efficient, and objective technology, it will facilitate the seamless comparison between studies. This will support the development of rehabilitation and surgical interventions with the aim to optimise patient outcomes. We aim to narratively review the accuracy and reliability of camera-based pose estimations of active shoulder range of motion, address challenges for standardisation, and offer insights on technology use, considering its strengths and limitations. This study lays the groundwork for integrating smartphone camera-based 2D-pose detection into routine clinical assessments, as well as for future research to validate 2D-pose methods.

## 2. Materials and Methods

This narrative review synthesises the existing literature, focussing on video-based pose technologies, or markerless motion capture, applied to clinical shoulder range of motion assessments. Rather than to systematically review the literature, as performed by Beshara et al. [19], our goal is to highlight recent findings and pinpoint directions for applying pose detection in pre/post-surgical shoulder care. We have not included the literature related to robot-assisted devices that can also assess upper limb range of motion and function (e.g., [20,21]) as the focus of this narrative review is on self-initiated active shoulder range of motion. By summarising and discussing the relevant literature, this review aims to contribute insights into the potential applications and advancements of 2D-pose technologies in the context of shoulder surgery and post-surgical shoulder rehabilitation.

To better understand the findings of this narrative review, it is important to briefly cover some foundational topics. The next sections will discuss the challenges posed by the shoulder joint’s large range of motion in assessments, the shoulder movements that are typically assessed in the clinical setting, the role of deep learning in identifying anatomical landmarks from video data, and the various video-based and depth sensor techniques commonly used for pose estimation in the literature.

### 2.1. Large Degree of Freedom of the Shoulder Joint Challenges Range of Motion Assessment

The human shoulder stands out as the most complex joint in the body, requiring coordinated movements across multiple shoulder girdle joints to align the numerous degrees of freedom required for hand positioning. Complex interactions between the scapulothoracic, acromioclavicular, sternoclavicular, and glenohumeral joints (Figure 2), coupled with relative thorax-to-pelvis movements, contribute to tasks like reaching.

A typical clinical assessment includes various shoulder movements to establish overall shoulder function, which is important for daily activities and self-care (Figure 3). These movements include frontal plane abduction, scapular plane abduction, sagittal plane forward flexion and extension, external rotation with elbows at the sides at 90° (external rotation in position 1), external and internal rotation with the shoulder and elbow at 90° abduction, and functional internal rotation to see how far up the hand can reach the back of the spine.

Clinically, range of motion is quantified using a universal goniometer, but due to time constraints, it is often assessed visually. The main issues with these methods are accuracy and between- and within-rater reliability (see introduction), causing variations in shoulder functional performance due to measurement techniques rather than actual function changes. Although a surgeon or therapist can get a good ‘feel’ of overall shoulder function by observing patient compensation strategies, these are challenging to quantify as multiple joints may be involved. A video captures the actual movements, but even with research-grade 3D-motion capture, quantifying these movements is challenging [23], let alone with video-based pose estimation methods [24]. One significant limitation of video-based pose estimation is its inability to detect the scapula orientation, hidden beneath skin and clothing. Consequently, current video pose methods are restricted to assessing the relative motion between the thorax and the upper arm (e.g., [18]). Despite this limitation, it aligns with existing clinical approaches relying on the universal goniometer and visual estimation that also quantify the thoracohumeral range of motion.

### 2.2. Deep Learning Technologies to Identify Body Landmarks from Video

Achieving an accurate identification of key body landmarks is paramount with any method estimating shoulder range of motion. The precision and consistency in identifying these landmarks, therefore, play an important role in ensuring accurate estimations of shoulder range of motion. The lack of reliability observed in visual and goniometric methods can likely be attributed to the variations in positional identification between and within clinicians [14,15,16]. Recent advancements in human pose estimation from images and videos, in automating and standardising the identification of body landmarks essential for precise shoulder range of motion calculations, have been made [25,26,27,28,29,30].

Human pose estimation actively identifies human figures from images and videos, discerning their posture through estimated joint locations via deep learning models [31]. This process is derived from annotated data sets, typically gathered from images/videos of humans in their surrounding or in motion capture studios (Table 1), to train machine and/or deep learning models [26]. Wang et al. [30] describes numerous open-source data sets. The widespread availability of these data sets has significantly advanced the science behind human pose estimation. However, the variation in training data sets, including methods and diverse annotated skeletal models derived from different manual, 3D motion capture marker placements, and 3D markerless motion capture (Table 1), may result in variations in distinct pinpointed locations of similar body landmarks relevant for shoulder range of motion. While human pose detection can automate the identification of key body landmarks, the use of different annotated models may impact the agreement of shoulder range of motion between different pose estimation algorithms (see Appendix A for detailed description of pose estimation algorithms). Therefore, understanding the specifics of the underlying training data set and the associated joint definitions is crucial for establishing trust in these artificial intelligence systems. Efforts have been made to combine datasets to enhance the robustness of pose estimation methods, aiming to unify the approach and address potential variations stemming from diverse training sources [32,33].

In general, two fundamental pose estimation methods exist: bottom-up and top-down. As outlined by viso.ai [34], bottom-up methods are initiated by estimating individual body joints and subsequently organising them to construct distinct poses (Figure 4). Conversely, top-down methods start by employing a person detector, identified with a bounding box, and then estimate body joints within those detected regions. While top-down methods might be better at localising the key points, bottom-up approaches offer superior real-time feedback [35], providing a quick assessment of its performance. It is noteworthy that most open-source pose estimations adopt the bottom-up approach; see the Appendix A for overview.

It is unclear which method suits shoulder range of motion assessment best. To quantify shoulder range of motion, a thorax reference orientation and an upper arm reference orientation need to be established from key landmarks. The relative angle between the two defines the shoulder angle (Figure 5). The shoulder angle is, therefore, dependent on the pose estimation methods and which body landmarks are included in the model. See below for further discussion.

### 2.3. Video-Based and Depth Sensor Technologies for Pose Estimation

The accuracy of shoulder range of motion estimates is linked to the employed technology as depicted in Figure 1. Different camera technologies (single/multiple) and various deep learning-based algorithms (see Appendix A) can be utilised to estimate the pose of a person. This estimation is based on anatomical landmarks extracted from video data, leading to diverse 2D or 3D estimates of the body landmarks. Notably, 3D pose-based body landmarks likely yield more accurate shoulder range of motion estimates compared to their 2D-pose counterparts; however, a direct comparison between the two have not been made. The key factor contributing to this difference is the potential for projection or parallax error onto the 2D camera plane [18]. This is crucial for the complex shoulder joint, where movements are possible beyond a single plane. Shoulder abduction movements, such as those in the scapular plane, as opposed to the frontal plane, for instances where the patient is not directly facing the camera, can introduce projection errors, leading to inaccurate shoulder range of motion estimates [18]. Both 2D-pose and 3D-pose methods determine the relative angle between the thorax and upper arm. However, 3D-pose methods have the potential advantage of correcting any misalignment of the participant relative to the video camera, as they provide a 3D orientation estimate of the thorax and can assess the plane of shoulder movements relative to the thorax. Despite the shoulder joint movement complexity, simple 2D-pose-based shoulder range of motion captured using a smartphone or single camera has the potential advantage of easy setup and data processing, making it, likely, a more time efficient option.

## 3. Results

### 3.1. Literature Overview of Studies That Used a Camera with Infrared/Near-Infrared (LiDAR) Emitter and Receiver

The Kinect (or Azure) camera, coupled with the Microsoft Software Developer Kit (SDK), has been widely used due to its general availability and cost-effectiveness for a 3D-pose markerless system. Numerous papers have assessed its performance in evaluating clinical active shoulder range of motion tests, comparing it with various other methods, such as visual estimation [36,37,38], universal goniometer [36,38,39,40,41,42,43], screen-based goniometer [44,45,46,47], and different 3D motion capture kinematical models [37,40,42,48,49,50,51,52,53], across individuals with [38,41,44,47] and without shoulder issues [36,37,39,40,42,43,45,46,48,49,50,51,52,53]. See Appendix B Table A1 for an overview of the included studies. This has resulted in a considerable variation in accuracy across studies utilising the Kinect camera or other similar cameras systems [42,43]. For instance, [45,47] established a close relation between markerless and screen based goniometry, whereas [38] observed a poor to moderate agreement between the universal goniometer and 3D-pose markerless system. The differences can be attributed to varying gold standards, movement protocols (static/dynamic), and distances from the cameras, with the accuracy diminishing as the distance increases, inter-operator variability (using universal goniometer), and study sample size [19]. Notably, what is considered as the gold standard emerges as a critical factor; for instance, the variation between the 3D-pose markerless system and gold standard universal goniometer method may be influenced by the inherent variability within and between raters of the latter [13,14,15,16]. This is underscored by the findings showing lower differences in test–retest values for the 3D-pose markerless system, compared to the goniometer-based methods [38,51]. Overall, the Kinect camera demonstrates a comparable accuracy in estimating shoulder range of motion against the referenced gold standards, exhibiting minimal bias (Table A1). It also exhibits good repeatability within and between raters in 3D-pose markerless systems, as highlighted in a systematic (see [19,54]) and scoping review [55] on the topic.

While 3D motion capture is potentially a superior gold standard compared to the universal goniometer, it comes with its own limitations. The markers used are skin-based and may not always accurately represent the underlying bony segments due to the relative movement between the skin and bone during motion. Despite this, 3D motion capture is considered the gold standard in kinematic analysis, ranking below 3D fluoroscopy (Figure 1). Several studies have compared 3D-pose markerless systems, such as the Kinect camera, against 3D motion capture [37,40,42,48,49,50,51,52,53]. However, the interpretation of these studies is hindered by the use of different marker sets supporting different models for extracting shoulder range of motion from 3D motion capture data. The choice of the 3D motion capture kinematical model significantly influences shoulder range of motion outcomes, leading to between-study variations. The studies employed three main 3D motion capture methods for comparison: (1) angle between lines connecting critical landmarks that reflect the thorax and upper arm (vector-based) [51,53]; (2) Plugin Gait Vicon model [37,42,49,52]; and (3) kinematic model based on the recommendation of the International Society of Biomechanics [12,40,50]. The impact of these kinematical models on outcomes remains unclear.

Consequently, study findings exhibit variability. Vector-based studies showed a good consistency of 3D-pose with 3D motion capture for abduction and flexion. However, differences in the abduction bias were reported [51,53], likely due to the variation in anatomical segment vector definitions. Plugin Gait studies [37,42,49,52] observed poor accuracy for internal and external shoulder rotation, contrasting the findings from Wilson et al. [37]. Flexion/extension and abduction/adduction demonstrated moderate relations between 3D-pose based and 3D motion capture [49], conflicting with certain studies [37]. The differences between the minimal and maximal range of motion for flexion and external rotation were similar between systems [52], although a bias interpretation is challenging as the starting angle was subtracted. The study conducted using an iPad camera combined with the iPad’s LiDAR system [42] reported a low consistency between 3D-pose (based on Apple vision SDK) and Plugin Gait shoulder angles for flexion and abduction, possibly due to their focus on end range of motion. For instance, at shoulder end of range, arm movements are potentially compounded by movements from thorax extension that are potentially not detected by the pose-based model. Lastly, two studies comparing the Kinect camera with the ISB-based 3D motion capture [40,50] revealed a low bias for flexion and extension but poor external rotation agreement [40] and a large bias for abduction [50] and moderate agreement for flexion, abduction, external rotation in position 1, and internal rotation [50].

In summary, 3D-pose is a promising technology that can be used to objectively quantify shoulder range of motion. However, it appears to perform less favourably when compared to 3D motion capture than when compared to goniometric-based measurements. The use of different 3D motion capture methods, the unclear derivation of shoulder angles from the relative 3D orientation between the thorax and upper arm, and different movement protocols contribute to the complexity of outcomes observed between these studies.

### 3.2. Literature Overview of Studies That Used Single Camera Video-Based Technologies for Pose Estimation

Limited studies have inquired the performance of single camera 2D-pose-based shoulder range of motion. The use of varying gold standards and diverse deep learning models estimating body landmark positions is creating some interpretative challenges. Notably, all studies focused on participants without shoulder issues. See Appendix B Table A2 for an overview of the included studies.

In evaluating the performance of different 2D-pose systems, OpenPose-derived abduction shoulder range of motion demonstrated good results when compared to image-based goniometry [56] and the referenced data sets [57]. Similarly, MediaPipe, coupled with in-house machine learning, exhibited a high consistency in shoulder abduction compared to the universal goniometer as reference [58]. In comparison with Kinect-based shoulder range of motion, Wazir et al. [59] reported good performance for abduction, flexion and external rotation using BlazePose; and the VGG-16 neural network showed a strong performance for abduction and adduction [60]. However, these studies [59,60] involved a limited number of participants. When compared to 3D motion capture, Clemente et al., [61] found good agreement in shoulder abduction, yet the occlusion of some 2D-pose landmarks based on MediaPipe hindered shoulder flexion performance in sitting contexts. Lastly, van den Hoorn et al. [18] compared shoulder range of motion derived using Apple vision, implementing a phone camera, against 3D motion capture with ISB recommendations, observing excellent consistencies in abduction, flexion, and extension but a poor performance in adduction. Despite a good consistency, 2D-pose exhibited some bias with 3D-motion capture [18].

In summary, single camera 2D-pose-based methods emerge as a promising technology, showcasing the favourable comparison against various gold standards in limited studies. However, the interpretation of the findings from these studies is complicated by the utilisation of different deep learning technologies and diverse methods for establishing a thorax reference, potentially introducing shoulder angle bias variations [18].

## 4. Discussion

Despite the growing interest in pose-based methods for assessing shoulder range of motion, there is significant variation in the reported accuracy across the included studies. This inconsistency stems largely from the diverse gold standards and movement protocols used to determine the accuracy of pose-based methods. The lack of standardised approaches for determining the accuracy of pose estimation methods creates challenges in comparing results. For instance, studies differ in whether they use static or dynamic protocols, which reference standards they employ (e.g., 3D motion capture, clinical goniometer, screen based clinometry, etc.), and how they define anatomical frames. These factors contribute to the observed variability, making it difficult to generalise findings. Without consistent validation frameworks, it remains challenging to establish clear benchmarks for the performance of pose technologies.

A critical challenge lies in the definition of body landmarks comprising the thorax for both 3D- and 2D-pose methods. This results in a loosely defined thorax, encompassing the entire spine region. The sparsity of landmarks in this area poses a concern, potentially leaving undetected movements between shoulder and hip landmarks, leading to an inaccurate thorax reference orientation. If not corrected, this could contribute to an overestimation of shoulder range of motion, especially when arm movement range is compounded by thoracic compensation/contribution. This consideration is crucial when optimising deep learning models for shoulder range of motion, given the likelihood of patients adopting compensatory movement strategies. Therefore, it is critical to validate methods in clinical populations. As highlighted earlier, movements outside the camera plane in single camera 2D-pose systems are susceptible to parallax errors [18]. Certain studies noted a bias against the gold standard while demonstrating a high consistency, emphasising the critical point that methods should not be used interchangeably if they do not agree, even in instances of high consistency.

### 4.1. Direction for Future Research

Defining the shoulder angle involves determining the relative angle between the thorax and the upper arm in any given method. The accuracy of the shoulder angle is influenced by how the thorax is defined, introducing more or less bias when compared to a gold standard [18]. In the context of pose estimation, an ideal thorax reference should rely on centrally detected body landmarks, excluding ipsilateral body landmarks (Figure 5). Pose models should be trained to identify landmarks that compensate for the sparsity of spinal landmarks common in most pose models. This would enhance the definition of the thorax reference system and improve the compensation for thoracic movement during arm elevation.

The biomechanical model for markerless motion capture follows ISB recommendations for 3D-motion capture and is relatively simple in nature. For arm elevation-type movements, it includes two landmarks defining the sternum (approximating the sternal notch and xiphoid processes) and two landmarks defining the upper arm (estimated elbow and shoulder joint). This model would improve the compensation for thoracic contribution, compared to models with sparse thorax definitions. It could assess the common shoulder elevation compensation by determining the angle between the sternal notch and the estimated glenohumeral joint, and the amount of thoracic contribution to arm elevation via the thorax (estimated sternum) orientation to the vertical.

For external and internal rotation movements (external rotation in position 1 or 2, Figure 3), the lower arm needs to be included in the model as upper arm rotations cannot be observed in pose estimation. When viewed from the front, single camera-based pose estimation is limited to the video plane projection-based estimation of external rotation. For example, when the projected distances between wrist and elbow joints coincide, the external rotation is 0°, and the distance increases with an increasing external rotation angle according to the following equation: ${\mathrm{ER}1=\sin}^{-1}\frac{p}{L}$. where ER1 is the external rotation angle in position 1, p is the 2D-projected distance between the elbow and wrist, and L is the estimated lower arm length, which needs to be estimated from the contralateral lower arm. A multiple camera setup (see Section 4.2 *How to estimate pose?*) would avoid relying on joint projection-based angle estimation and likely improve accuracy, especially with small or large external rotation angles. The external/internal rotation in position 2 needs to be estimated and viewed from the side and is defined by the relative angle between the lower arm and thorax.

For functional internal rotation movement (Figure 3), estimating the actual internal rotation angle based on a single camera pose estimation is challenging. Instead of angle, the reached position of the wrist relative to the spinal length could be used as a proxy, as is performed clinically.

The validation should include shoulder patients with varying pathologies before and/or after surgery. Active clinical shoulder range of motion tests should be recorded over time, throughout the full available range of motion, using multiple cameras strategically placed to view the participant directly from the front and angled from the sides, with appropriate validation checkerboards for post-data collection video camera position calibration. Time and positionally synced gold standard 3D motion capture should adhere to ISB recommendations to ensure consistency across studies and facilitate comparisons between different research findings. The deidentified data should be securely uploaded to dedicated servers, enabling the future validation of pose estimation optimised for this specific patient population.

Utilising current open-source pose estimation algorithms, further training should be focused on and optimised for shoulder assessment in patient populations, whether for single or multiple video camera applications. The goal is to enhance patient outcomes by establishing a standardised method and software that clinicians can freely use. This technology should be user-friendly (e.g., a phone app), time-efficient, and objective/valid for estimating shoulder range of motion. Standardisation allows for the comparison between studies, contributing to the development of new rehabilitation and surgical interventions. Furthermore, shoulder movement types can be classified for the ease of data processing and the use of single file recording.

### 4.2. How to Estimate Pose?

When employing single camera-based pose estimation, various factors in mitigating parallax error and optimising the use of available video resolutions need to be taken into account. Furthermore, understanding how the machine/deep learning models were trained plays an important role in the setup process. For instance, if the pose model is trained on the vision of the whole person, having the entire patient in view will optimise the confidence of identified landmarks. For the optimal setup of single camera-based 2D-pose methods, see Table 2.

Alternatively, a notable open-source software called OpenCap (version 4.2.2, Neuromuscular Biomechanics Laboratory, Stanford University, Stanford, CA, USA) [62] is available, utilising a multiple-camera setup with a minimum of two cameras and combining OpenPose and High-ResolutionNet pose algorithms. Although it demands more time for setup, OpenCap has the potential to accurately objectify shoulder range of motion. The positions of the cameras require calibration using a standard checker box pattern that can be permanently mounted on a wall visible to both cameras. A stable and fast internet connection is essential as pose estimation data processing occurs in the cloud, providing results directly. However, this method needs to be validated for shoulder applications and currently adopts a broad definition of the thorax, determined by shoulder and hip body landmarks. It is important to note that thoracic compensations compounding arm elevation may lead to an overestimation of shoulder range of motion.

## 5. Conclusions

3D motion capture based on ISB recommendations remains the gold standard for validating pose estimation technologies. The ongoing advancements in LiDAR/depth sensing technology, smartphone cameras, and machine learning algorithms contribute to the promising results in mobile phone camera-based pose estimations of shoulder range of motion within the clinical setting. However, it is imperative to exercise care as smartphone-based cameras are optimal for evaluating movements in a 2D plane—specifically, perpendicular to the camera’s position/orientation. Limitations arise when capturing rotational movements and forward flexion, as thoracic kyphosis and thoracic-related movements are not well captured and may give erroneous results. The camera positioning, orientation, and distance need to be optimised and the need to adhere to standardised protocols when using mobile phone camera-based methods to assess shoulder range of motion in the clinic is emphasised.

## Figures and Tables

**Figure 1 sensors-25-00667-f001:**
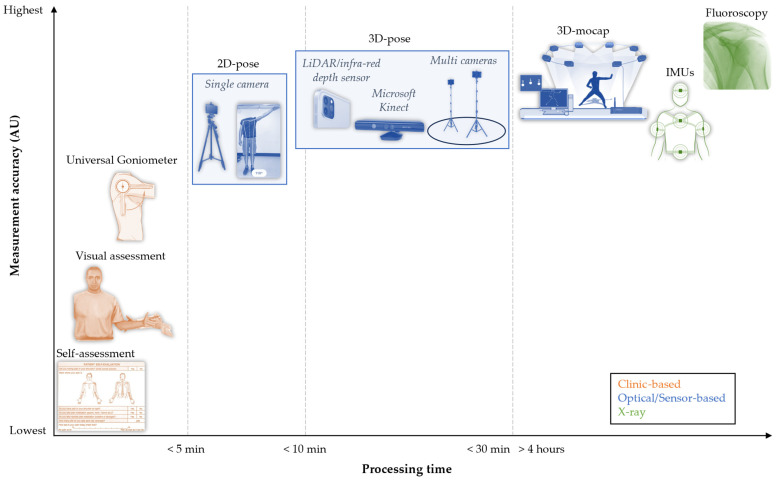
Speculative measurement accuracy vs. processing time of several methods to estimate shoulder range of motion. Clinical practice currently utilises diverse pose estimation tools, such as self-assessment, visual-assessment, and the universal goniometer (in orange). Within clinical settings, potentially more precise and consistent video/sensor-based methods (in blue), such as 2D-pose- and 3D-pose-based techniques, hold the potential for an enhanced accuracy of shoulder range of motion estimates. 2D-pose methods are based on a single camera, whereas 3D-pose methods can be either based on a single camera with an infrared, or a near-infrared Light Detecting and Ranging (LiDAR) depth sensor or multiple camera setups. 2D- and 3D-pose methods are coupled with machine/deep learning models that estimate the key body landmarks for estimating shoulder range of motion. In the realm of biomechanics, Inertial Measurement Units (IMUs) provide the orientation information; however, 3D motion capture is considered the gold standard. It surpasses the accuracy of other methods but requires trained staff to operate and process the data. 3D-fluoroscopy is ranked the highest (in green), but this approach involves the exposure to harmful radiation and necessitates the fitting of bony shapes, which consumes time.

**Figure 2 sensors-25-00667-f002:**
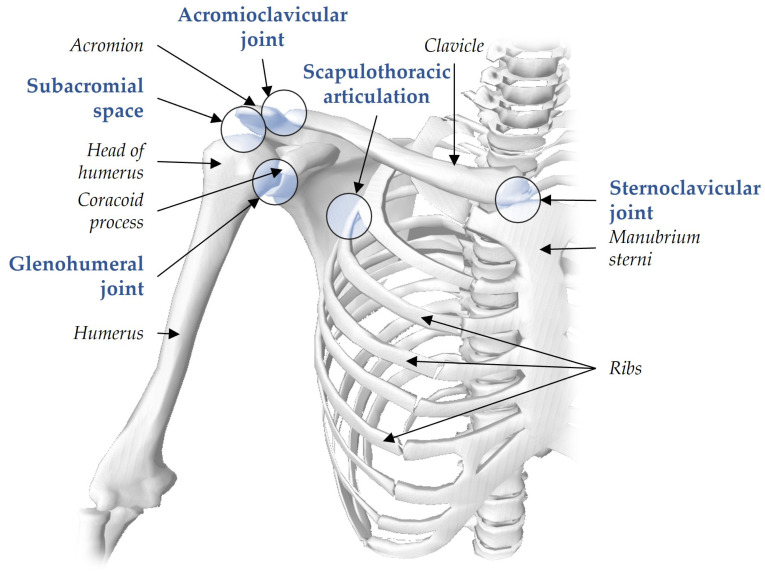
Anatomical illustration of the shoulder joint system. The shoulder girdle comprises three true joints and two functional joints, whose coordinated muscle activation enables its wide range of motion. The true anatomical joints are the sternoclavicular, acromioclavicular, and glenohumeral joints. The functional joints include the subacromial space, which facilitates the smooth gliding between the acromion and the rotator cuff, via bursae, and the scapulothoracic articulation, allowing the scapula to glide on the chest wall. This joint organisation underpins the shoulder’s remarkable mobility. Figure from OpenSim (version 4.4) [22].

**Figure 3 sensors-25-00667-f003:**
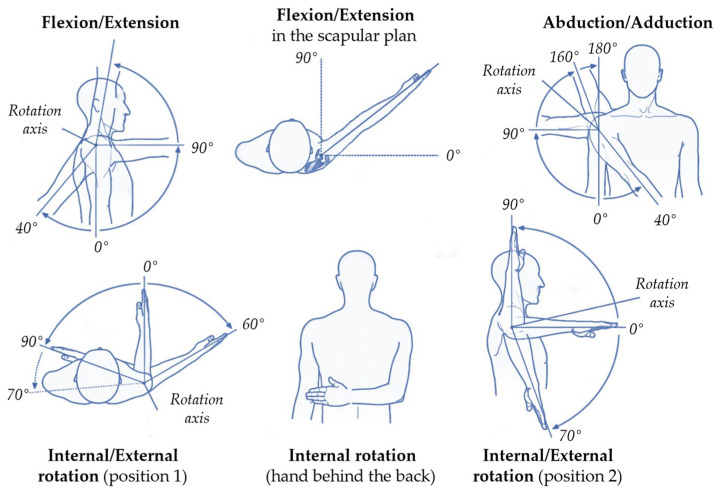
Active shoulder range of motion assessment. Illustration of shoulder movements commonly evaluated in clinical assessments.

**Figure 4 sensors-25-00667-f004:**
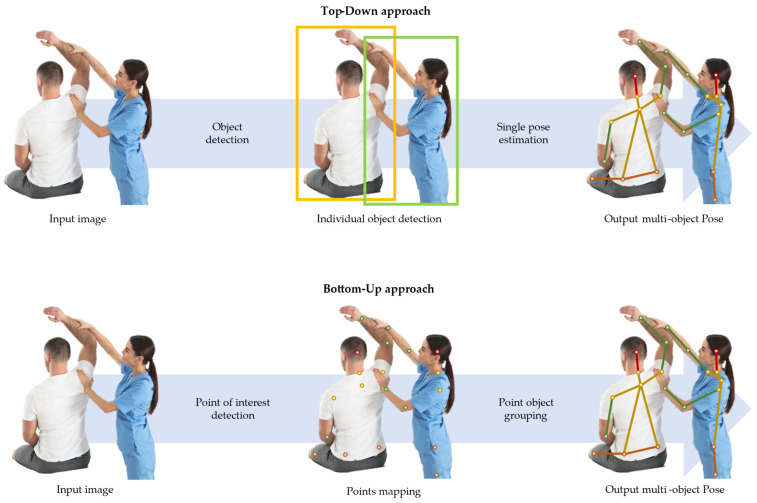
Simple illustration of the two basic approaches of machine/deep learning-based pose estimation. The top row depicts the top-down approach; first, persons are detected and bounded, then within the bounded person, key body landmarks are identified. The bottom row depicts the bottom-up approach; the individual’s body joint positions are estimated first and then subsequently organised to construct distinct poses.

**Figure 5 sensors-25-00667-f005:**
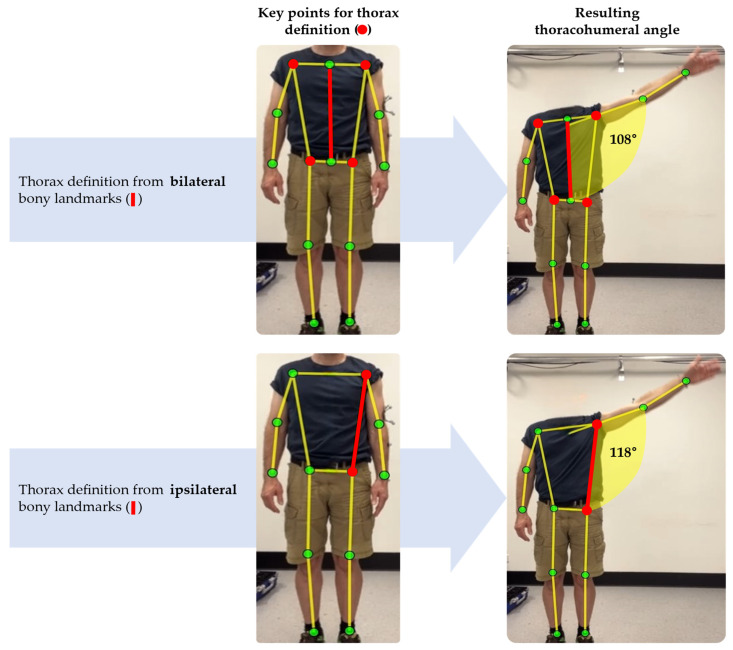
Impact of anatomical frame definition of the thorax (red lines) on estimated thoracohumeral angle. Both red and green dots represent identified body landmarks. For both rows, the thorax is loosely defined by the shoulder and hip markers and the upper arm is defined by the shoulder and elbow key points. The top row depicts the thorax definition based on the average position of the bi-lateral shoulder and hip markers (red dots), and the resulting thorax orientation is highlighted by the vertical red line. The bottom row depicts the thorax definition based on the ipsilateral shoulder and hip markers (red dots); consequently, the thorax orientation estimate (red line) is angled relative to the thorax definition of the top row, and the thoracohumeral angle will be biased and is estimated to be larger in the bottom row than the top row.

**Table 1 sensors-25-00667-t001:** Overview of annotated data sets used to train machine/deep learning algorithms for estimating human pose.

Dataset Name	Number of Images/Videos	Image Context	Annotation Method	Whole/Part Subject	Open Access	Website
LSP (Leeds Sports Pose)	~20 K	Sports activities (e.g., running, jumping)	Manually (AMT)	Whole person	Yes	https://github.com/axelcarlier/lsp
Max Panck Instutut Informatik (MPII) Human Pose	~25 K	Indoor environments (e.g., homes, offices)	Manually (AMT)	Whole person	Yes	http://human-pose.mpi-inf.mpg.de
Common Objects in Context (COCO)	~200 K	Diverse contexts (indoor, outdoor)	Manually (AMT)	Whole/part person	Yes	https://cocodataset.org/#home
Humans Interacting with Common Objects Detection (HICO-DET)	~47 K	Diverse contexts, including crowded scene	Manually	Whole/part person	Yes	https://www.v7labs.com/open-datasets/hico-det
Frames Labelled in Cinema (FLIC)	~5 K	frames of 30 movies	Manually (AMT)	Whole person	Yes	https://bensapp.github.io/flic-dataset.html
PoseTrack 2017	~500 videos	Various real-world settings (indoor, outdoor)	Manually (AMT)	Whole/part person	Yes	https://github.com/umariqb/PoseTrack-CVPR2017
PoseTrack 2018	~1.1 K videos	Various real-world settings (indoor, outdoor)	Manually (AMT)	Whole/part person	Yes	https://paperswithcode.com/dataset/posetrack
ImageNet Large Scale Visual Recognition Challenge (ILSVRC-2012)	~1 M	Diverse contexts	Class labels indicating objects/categories	N/A	Yes	https://www.image-net.org/challenges/LSVRC/2012/
CrowdPose	20 K	Crowded scenes	Manually (AMT)	Whole/part person	Yes	https://github.com/Jeff-sjtu/CrowdPose
Max Planck Institute for Informatics-3D Human Pose (MPI-INF-3DHP)	~3 K	Indoor environments (e.g., offices, labs)	3D-motion capture (markerless)	Whole person	Yes	https://vcai.mpi-inf.mpg.de/3dhp-dataset/
Human3.6 M	~3.6 M	Indoor environments (e.g., research labs)	3D-motion capture	Whole person	Yes	http://vision.imar.ro/human3.6m/description.php

AMT = Amazon Mechanical Turk, a crowdsourcing platform used to gather manual annotations for datasets. All websites accessed 1 June 2024.

**Table 2 sensors-25-00667-t002:** Optimal setup for 2D-pose-based estimation of shoulder range of motion.

Camera setup	Positioning of the camera: use tripod to mount the camera set at approximate shoulder height of the patient height can be fixed to limit setup time between patients the camera should not be angled up or down to limit parallax error a distance from the patient that ensures that the entire patient, including potential arm elevation, is just within view to ensure the background contrast is sufficient and plain for accurate feature extraction appropriate scene lighting to limit motion blur
Movements	During active clinical shoulder tests, movements should be as follows: performed at a moderate pace, ~10 s from the starting to ending position performed within the same plane as the camera
Video processing and storage	Post-video processing can be carried out using open-source software or with phone applications.
	Store video data for easy identification of patient and shoulder movement performed for future processing

## Data Availability

Not applicable.

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
