# Peer review of "The Future of Clinical Active Shoulder Range of Motion Assessment, Best Practice, and Its Challenges: Narrative Review"

_sensors, 2025, doi:10.3390/s25030667_

Round 1
Reviewer 1 Report
Comments and Suggestions for Authors
This review explores the use of video-based pose estimation and markerless motion capture technologies for assessing shoulder range of motion (ROM) in clinical settings. This is an interesting and relevant topic worthy of review.
The topic is original and relevant to the field, addressing a gap since no recent reviews on this topic have been identified.
This review summarizes the literature findings on new technologies for shoulder ROM assessments. However, an important set of references is missing: robotic systems used for assessing upper limb functions and ranges of movement for upper limb segments.
If the authors intentionally chose not to include such references in this review, this should be clearly stated in the text, as robotic devices for ROM assessment are among the novel and emerging technologies in this field.
I suggested the addition of a reference related to a novel robotic method for clinical ROM assessment. I am providing an exemplary reference, and even if such systems are not included in the review, there should be a mention of a few references of this type in the introduction and/or discussion:
Rodriguez-de-Pablo, C., Balasubramanian, S., Savić, A., Tomić, T.D., Konstantinović, L. and Keller, T., 2015, August. Validating ArmAssist Assessment as an outcome measure in upper-limb post-stroke telerehabilitation. In 2015 37th Annual International Conference of the IEEE Engineering in Medicine and Biology Society (EMBC) (pp. 4623-4626). IEEE.
Reviewer 2 Report
Comments and Suggestions for Authors
This paper reviewed the different approaches for assessing shoulder range of motion. As the authors stated, this is not a systematic review and It mainly focuses on more recent findings, especially sensor-based pose estimation methods. I have the following concerns/suggestions.
1. Section 2.1 should be expanded to go into more detail about the should range of motion assessment. For example, the authors should consider adding a figure to illustrate the complexity of shoulder joint biomechanics. Additionally, they should clearly define what the clinical needs are (e.g., which direction of motion should be examined), and what the current clinical standard practices are as well as their limitations.
2. In the discussion, the authors should consider providing suggestions for standardizing the biomechanical model used in 2D or 3D pose estimation methods for future studies (These suggestions should be based on the usefulness of the results in clinical practices). A standardized model would help comparisons across different studies.
Round 2
Reviewer 1 Report
Comments and Suggestions for Authors
Authors have successfully addressed my comments
Reviewer 2 Report
Comments and Suggestions for Authors
All my previous comments have been addressed.